# Banned by the law, practiced by the society: The study of factors associated with dowry payments among adolescent girls in Uttar Pradesh and Bihar, India

**Shobhit Srivastava**[1], **Shekhar Chauhan**[2], **Ratna Patel**[3], **Strong P. Marbaniang**[3], **Pradeep Kumar**[1]*, **Ronak Paul**[3], **Preeti Dhillon**[1]

**1** Department of Mathematical Demography & Statistics, International Institute for Population Sciences, Mumbai, India, **2** Department of Population Policies and Programmes, International Institute for Population Sciences, Mumbai, India, **3** Department of Public Health and Mortality Studies, International Institute for Population Sciences, Mumbai, India

* pradeepiips@yahoo.com

**Data Availability Statement:** The data can be found from the following link: https://dataverse.

## Abstract

### Background

Despite the prohibition by the law in 1961, dowry is widely prevalent in India. Dowry stems from the early concept of 'Stridhana,' in which gifts were given to the bride by her family to secure some personal wealth for her when she married. However, with the transition of time, the practice of dowry is becoming more common, and the demand for a higher dowry becomes a burden to the bride's family. Therefore, this study aimed to determine the factors associated with the practice of dowry in Bihar and Uttar Pradesh.

### Methods

We utilized information from 5206 married adolescent girls from the Understanding the lives of adolescents and young adults (UDAYA) project survey conducted in two Indian states, namely, Uttar Pradesh and Bihar. Dowry was the outcome variable of this study. Univariate, bivariate, and multivariate logistic regression analyses were performed to explore the factors associated with dowry payment during the marriage.

### Results

The study reveals that dowry is still prevalent in the state of Uttar Pradesh and Bihar. Also, the proportion of dowry varies by adolescent's age at marriage, spousal education, and household socioeconomic status. The likelihood of paid dowry was 48 percent significantly less likely (OR: 0.52; CI: 0.44–0.61) among adolescents who knew their husbands before marriage compared to those who do not know their husbands before marriage. Adolescents with age at marriage more than equal to legal age had higher odds to pay dowry (OR: 1.60; CI: 1.14–2.14) than their counterparts. Adolescents with mother's who had ten and above years of education, the likelihood of dowry was 33 percent less likely (OR: 0.67; CI: 0.45–

harvard.edu/dataset.xhtml?persistentId=doi:10.
7910/DVN/RRXQNT.

**Funding:** The author(s) received no specific
funding for this work.

**Competing interests:** The authors have declared
that no competing interests exist.

0.98) than their counterparts. Adolescents belonging to the richest households (OR: 1.48;
CI: 1.13–1.93) were more likely to make dowry payments than adolescents belonging to
poor households.

## Conclusion

Limitation of the dowry prohibition act is one of the causes of continued practices of dowry,
but major causes are deeply rooted in the social and cultural customs, which cannot be
changed only using laws. Our study suggests that only the socio-economic development of
women will not protect her from the dowry system, however higher dowry payment is more
likely among women from better socio-economic class.

## Introduction

The preponderance of dowry and bride-price practices is culturally driven and existed as a way
of marriage requirements [1]. In various traditional societies, the transfer of money or goods
accompanies the initiation of marriage. When made to groom families from the bride families,
such transfers are widely classified as dowry [2]. Historically, dowry served the fundamental
purpose of inheritance for women as men were thought to inherit the family property in the
Indian context [3]. Moreover, dowry has been seen as a way to compensate the groom's family
for the economic support they would provide to the new member of the family, i.e., the bride,
as women tend to have a small role in the market economy and are dependent on their hus-
bands [4]. The above interpretation holds true in the Indian scenario as historically; dowry has
been practiced in upper-caste families where women's economic opportunities are limited. In
the lower caste; where women are seen as economic contributors, the bride-price's custom was
more common [5]. However, dowry dynamics have been changing in recent times, and people
from the upper and lower caste are practicing dowry. Furthermore, recent studies noted that
the dowry system is prevalent across many cultures and is no longer is treated as a contribution
towards a suitable beginning of the practical life of newly married couples [6].

In India, dowry has been prevalent for ages. The custom of dowry in India is a deeply
rooted cultural phenomenon [7]. The concept of Sanskritization was proposed by eminent
sociologist Srinivas in 1952, and many communities that never took dowry before started prac-
ticing dowry probably due to the phenomenon of Sanskritization [8]. A study has noticed that
dowry is being practiced in about 93 percent of Indian marriage and is almost universal [9].
Not only this, but studies have also noted that dowry payments have increased manifolds in
India [10,11]. A clear explanation for rising dowry payments is the marriage squeeze [12].
However, in her study, Anderson refuted the claims of any association between marriage
squeeze and dowry payments [11]. Marriage squeeze depicts tightness of marriage market.
Chiplunkar and Weaver (2019) carefully documented the transition of dowry payments in
India using the 1999 wave of the ARIS-REDS data and test which theories about dowry infla-
tion are consistent with the data and which are not [13]. Chiplunkar and Weaver (2019) show
that the theory of sanskritization cannot explain dowry inflation. Similarly, they also find that
the REDS data offers limited support to the marriage squeeze hypothesis. Few researchers pos-
tulated the theory of 'sex ratios and dowry' whereby changes in sex ratios due to population
growth could alter dowry payments [8,14,15]. The spousal age gap difference remains a con-
cern as male marry at older ages than women, so when population grows, as was the case in
India in the 1950s and 1960s, there will be a surplus of women at marriageable ages relative to

men at marriageable ages. In the resulting "marriage squeeze", competition over relatively scarce grooms may cause an increase in dowry [2,16]. Contrary to these predictions, Chiplunkar and Weaver do not find that sex ratio in the marriage market is related to increases in the prevalence or size of dowry [13]. Instead, the "squeeze" appears to be relieved by changes in the age of marriage, with a smaller average age difference between brides and grooms [11]. Zhang & Chan (1999) utilizing 1989 Taiwan Women and Family Survey data of 25–60 years old women stated that dowry improves the bride's welfare in her family [17]. They further stated that dowry represents bequest by altruistic parents for a daughter which not only increases the wealth of new conjugal household but also enhances the bargaining power of the bride [17].

Researchers unanimously agreed that the issue of dowry could be associated with gender inequality and female deprivation [7,18]. Alfano (2017) argued that the presence of son preference resulting from deeply rooted attitudes that boys are more valuable than girls is mostly attributed to the dowry payments [19]. Alfano (2017) further stated that the economic intuition that sons are cheaper to raise than daughters stem from the dowry [19]. He opined that dowries increase the economic returns to sons and decrease the return to daughters [19]. Kumar (2020) is of the opinion that dowry prevails because of disempowerment of women, male dominance, and financial dependence on men [20]. He further stated that inability to give dowry causes victimization of brides; whereas, the glorification of dowry generates son preference leading to female feticide, sex-ratio imbalances, and gender inequality [20]. Prevailing son preference in Indian societies leads to female feticide so as to avoid the burden of dowry [21–23]. Bhalotra et al., (2020) found a positive relationship between gold prices and the value of dowry payments [21]. They stated that payment in gold is essential in Indian marriages and further validated that gold prices marked the financial burden of dowry [21]. Further, Bhalotra et al. (2020) evidently provided evidence that gold prices impact dowry value and, that parents react to unexpected increases in gold prices by committing girl abortion or neglecting girls in the first month of life, when neglect more easily translates into death [21].

Parents desire their daughters to marry educated men with urban jobs, because such men have higher and more certain incomes, which are not subject to climatic cycles and which are paid monthly, and because the wives of such men will be freed from the drudgery of rural work and will usually live apart from their parents-in-law. In a sellers' market, created by relative scarcity, there was no alternative but to offer a dowry with one's daughter [24]. A different notion was put forward by Bloch & Rao (2002), where they stated that husbands are more likely to beat their wives when the wife's family is rich because there are more resources to extract and the returns are greater [25]. A husband's greater satisfaction with the marriage, indicated by higher numbers of male children, reduces the probability of violence against women [25]. Thus, it is likely that aspects of violent behavior are strongly linked to economic incentives [25]. Previous research has documented numerous factors associated with the dowry, such as socio-economic factors of the families [26], failure of the government in curbing the practice of dowry [27], first-born gender in the family [28]. A study showed that increasing the returns to women's human capital could lead to the disappearance of marriage payments altogether [29]. Edlund (2006) hypothezing that rise in dowry payments in India has been associated with the disadvantaged position of women in the marriage market, has shown that in a much-used data set on dowry inflation, net dowries did not increase in the period after 1950 [30]. Moreover, the stagnation of net dowries after 1950 undermine claims that marriage market conditions for brides have worsened [30]. Answering the query of whether dowry is bequest or price, Arunachalam & Logam (2016) found that more than a quarter of marriages use dowry as bequests [31]. Arunachalam & Logam (2016) further noticed limited evidence on marriage squeeze as a factor for dowry [31].

It was around sixty years back when India enacted the Dowry Prohibition Act, 1961, to prohibit the giving or taking of the dowry. However, the act has been unsuccessful in curbing down the dowry's menace and failed in its basic fundamental of eliminating the demand for dowry [7]. The dowry is so profoundly entrenched that a way out seems a bit tedious task; even well-educated families begin saving wealth for their daughter after she is born in anticipation of the futuristic dowry payments [6]. Traditional marriage institutions affect the household's financial decisions and influence saving behaviour [28]. Despite acknowledging the problem of dowry widely, there is a paucity of empirical studies that systematically analyze the correlates of dowry among adolescent girls in recent times [13]. Given the growing concerns about the dowry's socioeconomic consequences, it is imperative to explore the correlates of dowry in India. Therefore, we have tried to examine the factors associated with dowry in India. This study captures data from adolescents aged 15–19 years of age. While examining the correlates of dowry among adolescent girls, this study contributes to the existing literature examining factors associated with dowry.

## Methods

### Data

This study's data came from Understanding the lives of adolescents and young adults (UDAYA) project survey conducted in two Indian states Uttar Pradesh and Bihar, in 2016 by the Population Council under the guidance of the Ministry of Health and Family Welfare, Government of India. The survey collected detailed information on family, media, community environment, assets acquired in adolescence, and quality of transitions to young adulthood indicators. A total of 150 primary sampling units (PSUs)—villages in rural areas and census wards in urban areas have been selected in the state in order to conduct interviews in the required number of households. The 150 PSUs were further divided equally into rural and urban areas, that is, 75 for rural respondents and 75 for urban respondents. Within each sampling domain, survey adopted a multi-stage systematic sampling design. The 2011 census list of villages and wards (each consisting of several census enumeration blocks [CEBs] of 100–200 households) served as the sampling frame for the selection of villages and wards in rural and urban areas, respectively. This list was stratified using four variables, namely, region, village/ward size, proportion of the population belonging to scheduled castes and scheduled tribes, and female literacy. The UDAYA provide the estimates for states as a whole as well as urban and rural areas of the states. The required sample for each sub-group of adolescents was determined at 920 younger boys, 2,350 older boys, 630 younger girls, 3,750 older girls, and 2,700 married girls in the state. The sample size for Uttar Pradesh and Bihar was 10,350 and 10,350 adolescents aged 10–19 years, respectively. The sample size for this study was 5,206 adolescent girls who were married at the time of the survey [32,33]. In the present study the unmarried boys and girls were dropped and only married adolescent girls were included in the sample. Fig 1 represents the sample selection procedure for the present study.

### Outcome variables

Dowry was the outcome variable of this study, which was binary. The question was framed as: "whether dowry paid at the time of marriage or later"? the response was coded as 0 means "no" and 1 means "yes." The variable measures the response of dowry if demanded during marriage or after marriage.

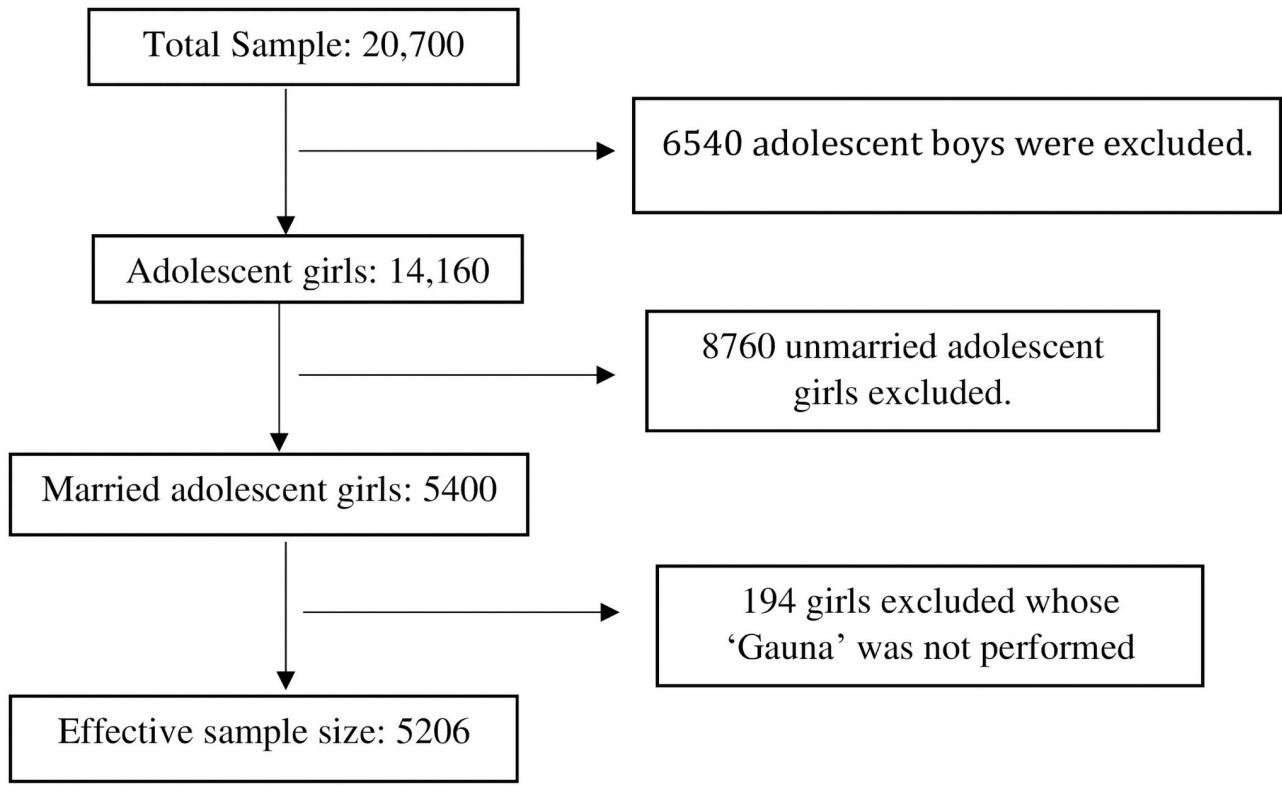

**Fig 1. Sample selection criterion for the present study.**

## Explanatory variables

1. Interaction with husband before marriage was named as "Husband known before marriage" and was recoded as not known and known.

2. Age at marriage was recoded as less than legal age (<18 years) and more than equals to legal age (≥18 years). The sample in 18 and above age category would be small as the data-set contained married adolescent's girl aged 15–19 years.

3. Spousal age gap was recoded wife older/almost the same age (wife older or one year younger than husband) and husband older (husband two or more years older than wife).

4. Spousal education recoded both not educated, only husband educated, only wife educated, and both educated.

5. Working status of the respondent was recoded as no and yes.

6. Whether vocation training was received or not by the respondent

7. Mother's education of the respondent was recoded as no education, 1–7, 8–9, and 10 and above years of education.

8. Land ownership among in-laws was coded as no and yes. The measurements about the land owned was not available in the data set.

9. Caste was recoded as Scheduled Caste and Scheduled Tribe (SC/ST) and non-SC/ST. The Scheduled Caste include a group of population which is socially segregated and financially/

economically by their low status as per Hindu caste hierarchy. The Scheduled Castes (SCs) and Scheduled Tribes (STs) are among the most disadvantaged socio-economic groups in India [34].

10. Religion was recoded as Hindu and non-Hindu.

11. Wealth index was recoded as poorest, poorer, middle, richer, and richest. The survey measured household economic status, using a wealth index composed of household asset data on ownership of selected durable goods, including means of transportation, as well as data on access to a number of amenities. The wealth index was constructed by allocating the following scores to a household's reported assets or amenities. Principal component analysis technique was used for creating the wealth index variable. The scores were divided into five quintiles using *xtile* command in Stata 14.

12. Residence was available in data as urban and rural.

13. Data were available for two states, i.e., Uttar Pradesh and Bihar, as the survey was conducted in these two states only.

## Statistical analysis

Univariate, bivariate, and multivariate logistic regression analysis [35] were performed to find the factors associated with dowry payment during marriage.

The equation for logistic distribution is given as:

$$ln\left(\frac{\pi}{1-\pi}\right) = \alpha + \beta_1 X_1 + \beta_2 X_2 + \beta_3 X_3 \ldots\ldots + \beta_n X_n$$

Where $\pi$ is the expected proportional response for the logistic model; $\beta_0, \ldots, \beta_M$ are the regression coefficient indicating the relative effect of a particular explanatory variable on the outcome. These coefficients change as per the context in the analysis in the study. *Svyset* command using Stata 14 was used to control for complex survey design. Additionally, individual weights were used to present the representative results. Further it was evident from the robustness check that logit model had a better fit. S4 and S5 Tables provide the summary statistics and correlation matrix along with plot of logistic predicted probabilities vs linear model (S1 Fig) and Plot of logistic predicted probabilities vs linear probability model (LPM) (S2 Fig).

Next, we check the stability of the regression coefficients and their sensitivity to selection bias using standard methods [36,37]. We obtain bias-adjusted coefficients and calculate the absolute deviation from the non-bias-adjusted regression estimates to understand the extent of bias. Further, we calculate Oster's δ, whose value higher than one would indicate that the regression coefficients are insensitive to omitted variable bias and variable selection bias [37]. All estimates were obtained with the assumption that the bias-adjusted model would explain 1.3 times variation in dowry payment status compared to the non-bias-adjusted model. The statistical analyses for coefficient stability check were performed using the *psacalc* command by estimating linear probability models in STATA [38].

## Results

The socio-demographic profile of the study population (married adolescents aged 15–19 years) is presented in Table 1. About 65 percent of adolescent girls did not know their husbands before marriage. Most husbands were older than their wives (91%) in the study population, and 64 percent of spouses (both) were educated. Only 11.7 percent of adolescent girls

**Table 1. Socio-demographic profile of married adolescents (15–19 years).**

| Variable | Sample | Percentage |
|---|---|---|
| **Husband known before marriage** | | |
| Known | 3368 | 64.7 |
| Not known | 1838 | 35.3 |
| **Age at marriage** | | |
| Less than legal age | 4151 | 79.7 |
| More than or equals to legal age | 1055 | 20.3 |
| **Spousal age gap** | | |
| Wife older/almost same age | 485 | 9.3 |
| Husband older | 4721 | 90.7 |
| **Spousal education** | | |
| Both not educated | 541 | 10.4 |
| Only husband educated | 710 | 13.6 |
| Only wife educated | 616 | 11.8 |
| Both educated | 3339 | 64.1 |
| **Working status** | | |
| No | 4596 | 88.3 |
| Yes | 610 | 11.7 |
| **Vocational training received** | | |
| Not received | 4400 | 84.5 |
| Received | 806 | 15.5 |
| **Mother education (in years)** | | |
| No education | 4319 | 83.0 |
| 1–7 | 449 | 8.6 |
| 8–9 | 241 | 4.6 |
| 10 and above | 196 | 3.8 |
| **In-laws land ownership** | | |
| No | 2999 | 57.6 |
| Yes | 2207 | 42.4 |
| **Caste** | | |
| SC/ST | 1543 | 29.7 |
| Non-SC/ST | 3663 | 70.4 |
| **Religion** | | |
| Hindu | 4296 | 82.5 |
| Non-Hindu | 910 | 17.5 |
| **Wealth index** | | |
| Poorest | 759 | 14.6 |
| Poorer | 1069 | 20.5 |
| Middle | 1222 | 23.5 |
| Richer | 1262 | 24.2 |
| Richest | 895 | 17.2 |
| **Place of residence** | | |
| Urban | 730 | 14.0 |
| Rural | 4476 | 86.0 |
| **State** | | |
| Uttar Pradesh | 3218 | 61.8 |
| Bihar | 1988 | 38.2 |
| **Total** | 5206 | 100.0 |

SC/ST: Scheduled Caste/Scheduled Tribe; Not legal age: Less than 18 years; Legal age: More than 18 years.

were working, and about 16 percent of adolescent girls received vocational training. Around 42 percent of girls' in-laws had land ownership. Nearly 30 percent of adolescents belonged to the SC/ST group, and most adolescents were Hindu (82.5%) and lived in rural areas (86%).

Table 2 depicts the distribution of adolescents who paid dowry by background characteristics. Overall, around 86 percent of adolescent girls reported that dowry was paid for their marriage. Bivariate results revealed that paid dowry was significantly higher among those who did not know their husbands before marriage (87.2%) than their counterparts (82.9%). It was more prevalent among those whose age at marriage was more than the legal age (89.1%). Similarly, dowry was more prevalent among married adolescent women whose husbands were older and higher if both husband and wife were educated. Interestingly, paid dowry was significantly higher among those who were not working (86%) and received vocational training (91%). Moreover, paid dowry was lower among the non-Hindu community (84.5%). Interestingly, the dowry was more prevalent in the richest households (88.7%). The rural-urban differential was observed for paid dowry. For instance, rural adolescents (86.8%) reported higher paid dowry than urban (78.7%) counterparts. S2 Table provides estimates for Uttar Pradesh and Bihar separately.

Estimates from logistic regression analysis for adolescents who paid dowry by important predictors were presented in Table 3. The likelihood of paid dowry was 48 percent significantly less likely (OR: 0.52; CI: 0.44–0.61) among adolescents who knew their husbands before marriage compared to those who do not know their husbands before marriage. Moreover, if adolescent girls who got marry after legal age (OR: 1.60; CI: 1.14–2.14) were 60 per cent more likely to pay dowry than their counterparts. The likelihood of paid dowry was 25 percent more likely among adolescents whose husband was older (OR: 1.25; CI: 1.03–1.67) than their counterparts. The odds of paid dowry were 39 percent, 47 percent, and 89 percent significantly more likely if only husband (OR: 1.39; CI: 1.05–1.83), only wife (OR: 1.47; CI: 1.11–1.96), and both were educated (OR: 1.89; CI: 1.48–2.4) respectively, than when both were not educated. Interestingly, if an adolescent's mother was having ten and above years of education, the likelihood of dowry was 33 percent less likely (OR: 0.67; CI: 0.45–0.98) than their counterparts. Wealth quintile has a positive relationship with adolescents who paid dowry for marriage. For instance, the odds of paid dowry were 33 percent, 39 percent, and 48 percent more likely among adolescents whose family gave dowry to marry them in middle (OR: 1.33; CI: 1.03–1.73), richer (OR: 1.39; CI: 1.06–1.83), and richest (OR: 1.48; CI: 1.07–2.05) families respectively compared to poorest counterparts. Moreover, the likelihood of paid dowry was 54 percent more likely in rural areas (OR: 1.54; CI: 1.28–1.86) than urban areas. Importantly, Bihar has higher odds for paid dowry (OR: 1.42; CI: 1.19–1.70) compared to Uttar Pradesh. Additionally, the estimates were provided for urban and rural place of residence as many covariates may vary by place of residence (S1 Table). Moreover, stepwise regression analysis was used to check for sensitivity bias (S3 Table).

Table 4 gives the results of the coefficient stability check of the explanatory variables of dowry payment among female adolescents. From the bias-adjusted estimates (see column 8), we observed that the multivariable association between husband familiar before marriage, age at marriage, spousal education, wealth index, residence and state with dowry payment is statistically significant (at 5% level) and lies in the same direction as the uncontrolled estimates. Moreover, from the difference shown in column 10, we can say that the bias-adjusted and non-bias-adjusted regression coefficients are similar. However, Oster's delta revealed that the statistically significant multivariable association of age at marriage and state with dowry payment suffers from omitted-variable and selection bias.

**Table 2. Percentage distribution of adolescents who paid dowry by background characteristics, 15–19 years.**

| Variable | Paid dowry (SD) | p<0.05 |
|---|---|---|
| **Husband known before marriage** | | * |
| Known | 82.9 (0.5) | |
| Not known | 87.2 (0.9) | |
| **Age at marriage** | | * |
| Less than legal age | 84.8 (0.5) | |
| More than or equals to legal age | 89.1 (1.1) | |
| **Spousal age gap** | | |
| Wife older/almost same age | 84.6 (1.8) | |
| Husband older | 85.8 (0.5) | |
| **Spousal education** | | * |
| Both not educated | 77.7 (1.5) | |
| Only husband educated | 82.4 (1.4) | |
| Only wife educated | 83.6 (1.4) | |
| Both educated | 88.0 (0.5) | |
| **Working status** | | * |
| No | 86.0 (0.5) | |
| Yes | 83.3 (1.5) | |
| **Vocational training received** | | * |
| Not received | 84.7 (0.5) | |
| Received | 91.1 (1.0) | |
| **Mother education (in years)** | | * |
| No education | 85.1 (0.5) | |
| 1–7 | 87.2 (1.6) | |
| 8–9 | 90.5 (1.8) | |
| 10 and above | 89.2 (2.1) | |
| **In-laws land ownership** | | |
| No | 84.4 (0.6) | |
| Yes | 87.4 (0.8) | |
| **Caste** | | * |
| SC/ST | 85.4 (0.9) | |
| Non-SC/ST | 85.8 (0.5) | |
| **Religion** | | * |
| Hindu | 85.9 (0.5) | |
| Non-Hindu | 84.5 (1.2) | |
| **Wealth index** | | * |
| Poorest | 81.6 (1.3) | |
| Poorer | 83.4 (1.1) | |
| Middle | 87.0 (0.9) | |
| Richer | 86.6 (0.9) | |
| Richest | 88.7 (1.1) | |
| **Place of residence** | | * |
| Urban | 78.7 (0.9) | |
| Rural | 86.8 (0.5) | |
| **State** | | * |
| Uttar Pradesh | 84.4 (0.8) | |
| Bihar | 87.8 (0.5) | |
| **Total** | 85.7 | |

*if p<0.05; SC/ST: Scheduled Caste/Scheduled Tribe; SD: Standard Deviation; Not legal age: Less than 18 years; Legal age: More than 18 years.

**Table 3. Logistic regression estimates for adolescents who paid dowry by background characteristics (15–19 years).**

| Variables | AOR (95% CI) |
|---|---|
| **Husband known before marriage** | |
| Known | 0.52*(0.44,0.61) |
| Not known | Ref. |
| **Age at marriage** | |
| Less than legal age | Ref. |
| More than or equals to legal age | 1.60* (1.14; 2.24) |
| **Spousal age gap** | |
| Wife older/almost same age | Ref. |
| Husband older | 1.25*(1.03,1.67) |
| **Spousal education** | |
| Both not educated | Ref. |
| Only husband educated | 1.39*(1.05,1.83) |
| Only wife educated | 1.47*(1.11,1.96) |
| Both educated | 1.89*(1.48,2.4) |
| **Working status** | |
| No | Ref. |
| Yes | 0.84(0.66,1.06) |
| **Vocational training received** | |
| Not received | Ref. |
| Received | 1.16(0.91,1.47) |
| **Mother education (in years)** | |
| No education | Ref. |
| 1–7 | 0.98(0.72,1.34) |
| 8–9 | 1.39(0.89,2.18) |
| 10 and above | 0.67*(0.45,0.98) |
| **In-laws land ownership** | |
| No | Ref. |
| Yes | 1.09(0.89,1.34) |
| **Caste** | |
| SC/ST | Ref. |
| Non-SC/ST | 1.09(0.9,1.31) |
| **Religion** | |
| Hindu | Ref. |
| Non-Hindu | 0.93(0.74,1.17) |
| **Wealth index** | |
| Poorest | Ref. |
| Poorer | 1.10(0.85,1.42) |
| Middle | 1.33*(1.03,1.73) |
| Richer | 1.39*(1.06,1.83) |
| Richest | 1.48*(1.07,2.05) |
| **Place of residence** | |
| Urban | Ref. |
| Rural | 1.54*(1.28,1.86) |
| **State** | |
| Uttar Pradesh | Ref. |
| Bihar | 1.42*(1.19,1.7) |

*if p<0.05, Ref: Reference; AOR: Adjusted Odds Ratio; CI: Confidence Interval; SC/ST: Scheduled Caste/Scheduled Tribe; Not legal age: Less than 18 years; Legal age: More than 18 years.

**Table 4. Coefficient stability results of the Linear probability model estimates for the multivariable association between dowry payment and explanatory characteristics.**

| Characteristics | Uncontrolled estimates | | | Controlled estimates | | | Bias-adjusted estimates | | Difference in | Degree of bias |
|---|---|---|---|---|---|---|---|---|---|---|
| | Coefficient | SE | $R^2$ | Coefficient | SE | $R^2$ | Coefficient | SE | Coefficients[(c)] | Delta (δ) |
| Husband known before marriage | -0.079* | (0.010) | 0.011 | -0.082* | (0.011) | 0.031 | -0.084* | (0.011) | 0.002 | 23.016 |
| Age at marriage | 0.036* | (0.014) | 0.001 | 0.047* | (0.014) | 0.031 | 0.051* | (0.013) | 0.004 | -13.895 |
| Spousal age gap | 0.025 | (0.019) | 0.000 | 0.030 | (0.019) | 0.031 | 0.031 | (0.016) | 0.001 | -22.415 |
| Spousal education | 0.026* | (0.004) | 0.007 | 0.025* | (0.005) | 0.031 | 0.024* | (0.005) | 0.001 | 5.842 |
| Working status | -0.046* | (0.015) | 0.002 | -0.027 | (0.016) | 0.031 | -0.020 | (0.017) | 0.007 | 3.824 |
| Vocational training period | 0.025 | (0.014) | 0.001 | 0.018 | (0.014) | 0.031 | 0.016 | (0.012) | 0.002 | 7.909 |
| Mother's education (in years) | 0.003 | (0.005) | 0.000 | -0.005 | (0.005) | 0.031 | -0.007 | (0.005) | 0.002 | -1.982 |
| In-laws land ownership | 0.037* | (0.010) | 0.002 | 0.005 | (0.012) | 0.031 | -0.010 | (0.015) | 0.015 | 0.332 |
| Caste | 0.021* | (0.011) | 0.001 | 0.008 | (0.011) | 0.031 | 0.003 | (0.013) | 0.005 | 1.505 |
| Religion | -0.028* | (0.013) | 0.001 | -0.008 | (0.014) | 0.031 | 0.000 | (0.016) | 0.008 | 1.048 |
| Wealth Index | 0.011* | (0.004) | 0.002 | 0.014* | (0.004) | 0.031 | 0.015* | (0.005) | 0.001 | 12.206 |
| Residence | 0.045* | (0.010) | 0.004 | 0.053* | (0.011) | 0.031 | 0.057* | (0.013) | 0.004 | 14.698 |
| State | 0.032* | (0.010) | 0.002 | 0.040* | (0.011) | 0.031 | 0.043* | (0.012) | 0.003 | -41.775 |
| Analytical sample size | **5,206** | | | **5,206** | | | **5,206** | | | |

Note–(1) * denotes p-value<0.05; (2) SE: Standard Error, $R^2$: Coefficient of determination of linear probability model;
[(c)]Absolute difference between the controlled and bias-adjusted coefficients;
(d) Value of δ<1 denotes biased coefficients.

## Discussion

The practice of dowry is widely prevalent in India [9], despite the prohibition by the law in 1961. Dowry stems from the early concept of 'Stridhana,' in which gifts were given to the bride by her family to secure some personal wealth for her when she married [39]. However, with the transition of time, dowry has become a common practice, and the demand for a higher dowry becomes a burden to the bride's family. Srinivasan (2005) describes that modern dowry comprises demands that include gold, cash, and consumer goods that far exceed what families can afford, exploiting its obligatory symbolic nature and the fact that they are gifts of love woman from her natal family [40]. This study aimed to determine the factors associated with the practice of dowry among adolescent girls. The study reveals that the practice of dowry is still prevalent and is influenced by many factors such as spousal age gap, spousal education, and household socioeconomic status.

Our study report that a girl knowing her husband before marriage is less likely to pay dowry than those not known about the husband. This may be because the information about knowing each other is much better in a love match [41], and the practice of dowry is almost non-existent in the case of love marriage [40,42]. Other studies mentioned that in an arranged marriage where information about each other is limited, the groom's quality is inferred through his education, associated with the dowry level [41]. The age of the bride is the main factor in marital negotiation, particularly in rural India. Our study found that a girl married to a husband older than her is more likely to pay dowry than an older girl or of similar age to her husband. One possible explanation by Maitra (2006) in his study on dowry inflation in India, argues that the excess supply of younger brides in the marriage market can leads to increase in dowry price when an older man marries the younger brides [43]. Another reason could be groom late age at marriage may be associated with pursuing of higher education [44]. Higher

groom education is often found to be associated with higher dowry; this is because due to the competition among the brides for a particular groom leads to offers of higher and higher prices of dowries [41]. Also, suppose a potential bride's cares about the qualities of the groom like commitment, sincerity, and loyalty which is important for a peaceful marriage, however if these qualities are unobservable and likely to be true, the brides may judge from the groom education as the signal of these qualities [45]. Hence, the bride's family is ready to pay more dowries for a more educated person, not for higher education, but the underlying desirable qualities signals [45].

However, our study reveals contrasting results as girl education does not impact reducing the amount of dowry paid. Dalmia & Lawrence (2005) while examined the continued prevalence of dowry system in India explained that the amount of dowry or money transfer from brides and their families to grooms and their families does not decrease with increasing bride's level of education [2]. Our results show that educated girls are more likely to pay dowry than the uneducated girls whose husband is also not educated. One possible explanation could be that the brides' education is a good indicator of her household wealth. Hence, higher education and higher dowry are effects of bride's household wealth [45]. Girl having a higher level of education tends to marry at a later age because they are more job aspirants than the lower educated girl [46]. The study of Dhamija & Chowdhury(2020) noted that a delay in marriage is associated with more education, low fertility, and possibly higher dowry for Indian women [47]. Findings by Field & Ambrus (2008) show that marriage opportunities curtail schooling investment suggest that the benefits to girls of delaying marriage come at a cost to the families, probably in higher dowry payments or less desirable spouses [48]. A more educated girl puts her parents in a difficult situation because it is very difficult to get a suitable boy for an educated girl. By virtue of her feminine status, a girl is expected to marry a man who should be in a better position and more educated than her. Drèze & Sen (1995) explained that if an educated girl marries a more educated boy, then the dowry payment will be more likely to increase with the groom's education [49]. As Mathew (1987) explained, the expected dowry's mean value increased with the prestige of the groom's education [50]. Foreign degrees drew the highest dowries, Ph.D. degree received the lowest than engineering and medical degrees. However, on the other way, in the case of a rift, a more educated bride is more likely to walk out of the marriage, and the groom is bearing a greater risk of separation. So, given grooms value marital life's stability or longevity, they will want to be paid a higher price for marrying with more educated brides as a premium for bearing the additional risk that such marriages entail [45].

The studies of Saroja & Chandrika (1991) found that as the bride, parental income increased, and dowry also increased [51]. This is consistent with our study's findings, where the girls from the wealthier family were more likely to pay the dowry at marriage. The possible reason for this result is that the higher the parents' income, the chance with which dowry demands can be agreed with ease, and more smoothly, the dowry payment can take place [52]. Our study shows that girls from Bihar were more likely to pay dowry than girls in Uttar Pradesh, this finding warrant qualitative study on dowry practices between these two states, because the information from the present study is not sufficient to draw a conclusion for this finding.

The study has some limitations as the study was conducted only in two Indian states, so the researchers could not establish a general conclusion from this study. Also, as our study in quantitative, we are unable to capture the individual social and cultural view point on dowry practice. Additionally, national representative data will be helpful for further study to understand the scenario of dowry practice in India, as because India is a country with diverse social and cultural practices, dowry will vary with respect to their cultural norms. Although the coefficient stability check revealed that the majority of the explanatory characteristics are

insensitive to omitted-variable and selection bias, the results for age at marriage and state need to be interpreted with caution.

## Conclusion

The study sought to explore the factors associated with the practice of dowry in Bihar and Uttar Pradesh. It is evident from this study that the practice of dowry is still widespread, and the results show that increasing age, education, and household economic status of girls are associated with the likelihood of high dowry payment. Limitation of the dowry prohibition act is one of the causes of continued practices of dowry [53]. However, significant causes are deeply rooted in social and cultural customs, which cannot be changed using laws. Our study suggests that socio-economic development of women will not protect her from the dowry system, however higher dowry payment is more likely among women from better socio-economic class.

## Supporting information

**S1 Fig. Plot of logistic predicted probabilities vs. linear model.**
(DOCX)

**S2 Fig. Plot of logistic predicted probabilities vs. LPM.**
(DOCX)

**S1 Table. Logistic regression estimates for adolescents who paid dowry by background characteristics (15–19 years).**
(DOCX)

**S2 Table. Percentage distribution of adolescents who paid dowry by region, 15–19 years.**
(DOCX)

**S3 Table. Stepwise logistic regression estimates for adolescents who paid dowry by background characteristics (15–19 years).**
(DOCX)

**S4 Table. Summary statistics for LPM.**
(DOCX)

**S5 Table. Correlation index for robustness check of LPM.**
(DOCX)

## Acknowledgments

Authors are thankful to Population Council, India for providing UDAYA data for research. This paper was written using data collected as part of Population Council's UDAYA study, which is funded by the Bill and Melinda Gates Foundation and the David and Lucile Packard Foundation.

## Author Contributions

**Conceptualization:** Shobhit Srivastava.

**Data curation:** Shobhit Srivastava, Pradeep Kumar.

**Formal analysis:** Shobhit Srivastava, Pradeep Kumar, Ronak Paul.

**Investigation:** Shobhit Srivastava, Pradeep Kumar.

**Methodology:** Pradeep Kumar.

**Resources:** Pradeep Kumar.

**Software:** Shobhit Srivastava, Pradeep Kumar.

**Supervision:** Preeti Dhillon.

**Validation:** Shobhit Srivastava, Pradeep Kumar, Preeti Dhillon.

**Writing – original draft:** Shekhar Chauhan, Ratna Patel, Strong P. Marbaniang.

**Writing – review & editing:** Preeti Dhillon.

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
