## [Decision Letter · Decision Letter 0]

10 May 2021

PONE-D-21-04546

Banned by the law, practiced by the society: The study of factors associated with dowry payments in Uttar Pradesh and Bihar, India

PLOS ONE

Dear Dr. Kumar,

Thank you for submitting your manuscript to PLOS ONE. After careful consideration, we feel that it has merit but does not fully meet PLOS ONE’s publication criteria as it currently stands. Therefore, we invite you to submit a revised version of the manuscript that addresses the points raised during the review process.

I sent this paper to 3 very knowledgeable referees who have all suggested "Major Revision". I have read this paper with interest and I concur with the comments made by the referees. I believe the paper will improve a lot if you are able to address the comments (to the extent possible) -- on exposition, sample selection, writing (all three agree on this), empirical (R1 and R2).

I do realize that it might be a lot of additional work so I will let you decide if you would like to re-submit, however, I hope you do send us the revised version. Please submit your revised manuscript by July 10, 2021. If you will need more time than this to complete your revisions, please reply to this message or contact the journal office at plosone@plos.org. Please include the following items when submitting your revised manuscript:

We look forward to receiving your revised manuscript.

Kind regards,

Nishith Prakash, Ph.D.

Academic Editor

PLOS ONE

2. Please amend either the title on the online submission form (via Edit Submission) or the title in the manuscript so that they are identical.

Reviewers' comments:

Reviewer's Responses to Questions

**Comments to the Author**

1. Is the manuscript technically sound, and do the data support the conclusions?

Reviewer #1: Partly

Reviewer #2: Partly

Reviewer #3: Partly

2. Has the statistical analysis been performed appropriately and rigorously? 

Reviewer #1: Yes

Reviewer #2: No

Reviewer #3: Yes

3. Have the authors made all data underlying the findings in their manuscript fully available?

Reviewer #1: Yes

Reviewer #2: Yes

Reviewer #3: Yes

4. Is the manuscript presented in an intelligible fashion and written in standard English?

Reviewer #1: No

Reviewer #2: Yes

Reviewer #3: No

5. Review Comments to the Author

Reviewer #1: Referee Report on "Banned by the law, practiced by the society: The study of factors associated with dowry payments in Uttar Pradesh and Bihar, India"

Summary & General Evaluation

Dowries are wealth transfers from the bride’s family to the groom or groom’s family paid at the time of the wedding. In India, the practice of dowry payment is highly prevalent and typically several times the yearly household income, despite being illegal since 1961. This paper studies the factors associated with a dowry payment using data from the Understanding the Lives of Adolescents and Young Adults (UDAYA) project, consisting of 5206 married adolescent girls from two states in India, Utter Pradesh, and Bihar.

Using logistic regression analysis, the authors document associations between the practice of dowry payment and various socio-economic factors such as the age of the girl, education of the girl, household socioeconomic status, spousal education and, spousal age gap, to name a few.

Let me start by saying that I find the exercise of establishing correlations between dowry payments and important socio-economic variables worthwhile, and the paper definitely furthers our understanding of the possible determinants of the decision to pay a dowry along with confirming some of the previous findings in the literature. However, in its current version, the paper could be significantly improved in terms analysis and writing. Below I summarize my main comments.

Main Comments

Data and Regression Analysis. The following are my specific concerns with the data and analysis in the paper (The main text from the article is in blue with quotations):

"The sample size for this study was 5,206 adolescent girls who were married at the time of the survey" – The original sample is 10,350 adolescents from Utter Pradesh and 10,350 adolescents from Bihar; from this, you select 5206 adolescents. Please explain the sample selection procedure.

"Wealth index was recoded as poorest, poorer, middle, richer, and richest. The survey measured household economic status, using a wealth index composed of household asset data on ownership of selected durable goods, including means of transportation, as well as data on access to a number of amenities. The wealth index was constructed by allocating" - Doing a Principal Component Analysis on the variables and, creating a wealth component would be preferred.

"This is consistent with our study's findings, where the girls from the wealthier family were more likely to pay the dowry at marriage." – I am not sure what variable are you using for natal family wealth?

Since your sample by deﬁnition only includes girls aged 15-19. The fraction of girls who are legally going to marry above the legal age of 18 is by construction going to be small. The authors should mention this point in the paper.

π needs to defined precisely in the specified logistic regression.

As a robustness check, the authors could see if these correlations hold true using a linear probability model instead of logistic regression.

Relation to existing works. The authors could have been more thorough in writing this paper, especially in citing relevant literature while explaining the main ﬁndings. There is extensive theoretical literature in economics on the emergence and the existence of dowries. In the introduction, while motivating the presence of dowries in societies like India, the authors cite some of the papers such as Anderson (2003) but miss out on some critical articles like Anderson and Bidner (2015); Botticini and Siow (2003).

Similarly, there is a growing empirical literature studying the determinants of dowry payments that have carefully analyzed questions related to this paper’s main ﬁndings. I recommend the authors carefully review the following articles:

Chiplunkar and Weaver (2019) carefully document the transition of dowry payments in India using the 1999 wave of the ARIS-REDS data and test which theories about dowry inﬂation are consistent with the data which are not. I highly recommend the authors to read this paper thoroughly. For example, the authors in the introduction talk about the Sanskritization theory, Chiplunkar and Weaver (2019) show that this theory cannot explain dowry inﬂation. Similarly, they also ﬁnd that the REDS data offers limited support to the marriage squeeze hypothesis.

Edlund (1999): The author also studies the hedonic regressions of dowry on bridal traits. However, she looks at actual magnitude dowry payments, different from this paper that looks at dowry payments on the extensive margin. A couple of sentences comparing results in this paper to yours will be beneﬁcial.

Arunachalam and Logan (2016) is also a related paper.

Exposition and takeaways. The discussion and the conclusion section need to significantly re-written for clarity. The authors make a series of claims in the discussion section that require a relevant citation. Similarly, the conclusion section can also be reworded in line with the main contribution of the paper. I list some of the specific instances below (The main text from the article is in blue with quotations):

"Researchers unanimously agreed that the problem lies with gender inequality and female deprivation at every stage" - This sentence needs to be reworded for clarity, and relevant literature that has documented the relationship between dowry payment and gender inequality at different stages of a woman’s life needs to be cited (Alfano, 2017; Bhalotra et al., 2020; Bloch and Rao, 2002; Zhang and Chan, 1999). Further, the claim that dowries are associated with female deprivation at every stage is not supported by the existing literature (Zhang and Chan, 1999).

"Despite acknowledging the problem of dowry widely, there is a paucity of empirical studies that systematically analyze the correlates of dowry among adolescent girls in recent times." - The authors need to cite relevant papers here.

"However, with the transition of time, the practice of dowry is becoming mandatory, and the demand for a higher dowry becomes a burden to the bride’s family." - There is a shift in dowries from a stridhan to a groom-price (Srinivas, 1984), but what is the evidence that the practice is becoming more mandatory?

"Finally, national representative data will be helpful for further study to understand the scenario of dowry practice in India, as because India is a country with diverse social and cultural practices, dowry will vary with respect to their cultural norms." - Rural Economic and Demographic Survey (REDS) of India is a nationally representative survey that contains dowry information.

"Instead, a massive social reform and action are urgently required to stop the practice and change their attitude about the system. A community-level approach is necessary to develop their level of understanding and awareness to understand the negative impact of such an evil custom. Simultaneously, it is necessary to restructure the existing dowry prohibition law to make it more effective. There are some unique and exceptional causes regarding dowry that need to be considered during policy development." - This goes well beyond the scope of the paper. The paper is documenting interesting correlations in a unique dataset between dowry payment and socio-economic characteristics. The conclusion should be about these associations.

Other comments

"However, in his study, Anderson refuted the claims of any association between marriage squeeze and dowry payments [11]." - This sentence has a notable typo; it should be in her study.

The paper in its current version has several typos and grammatical errors. The references section needs to be edited, too; please follow one reference style consistently throughout the paper.

Reviewer #2: Comments regarding three major areas - contribution, mechanisms and empirical analysis are attached. These are the key areas in this which this paper needs a significant improvement from its current draft.

Reviewer #3: The paper uses data on 5206 married adolescent girls from the Understanding the lives of adolescents and young adults (UDAYA) conducted in Uttar Pradesh and Bihar for studying correlates of dowry payment in India. Main findings are - dowry likelihood lower if husband is known to the female, if the adolescent’s mother has more than 10 years of education; dowry likelihood higher if the couple is more educated, girl above legal age, husband was older, wealthier families, and in rural areas.

Major comments

1) Definition of dowry and who answered the question matters: While the authors mention that their main variable of interest comes from what a household’s response is to the question – “whether dowry paid at the time of marriage or later?” – what is not clear is the inclusions in the term “dowry”. A clarity on this would be valuable for the readers. There are two potential measurement issues with this variable

o One, if perception of dowry (inclusions) and hence reporting varies by education or other economic correlates, then this can potentially contaminate the findings of the paper.

o Second, dowry is a sensitive issue and reporting might vary – although whether it varies along the dimensions that the authors study would need to be argues. If it is underreported by the same fraction by all groups then it does not matter. However, the rural-urban differential can vary because of reporting sensitivity too where urban households maybe aware of dowry prohibition act.

o Importantly, who answers the question is also important and implications of these both should be adequately discussed, even though I suppose addressing these issues is not feasible.

2) The authors mention – “Higher education is often found to be associated with higher dowry; this is because due to the competition among the brides for a particular groom leads to offers of higher and higher dowries” and cite a paper by Munshi (2017). But I am not certain if this is the only theoretical channel that should drive the effect of education on dowries. It is possible that more education leads to more awareness about evils of dowry and can potentially also lower the dowry payment? In general, it would be good to present a theoretical framework of why each result that the authors obtain can be justified theoretically (not a theoretical model, but channels are enough). At least, what are the hypothesis of the authors (based on different channels) should be mentioned before they go on to the empirical strategy and discuss the covariates they include. The current discussion in the results seems a bit superfluous and lacks conceptual clarity, hence the results do not seem to come together coherently.

3) While discussing the results the authors discuss results from previous studies which examine both probability of dowry payment and amount of dowry paid – it would be good for a reader to clearly differentiate between these two types of studies.

4) The authors also make an argument on page 13 about girls from Bihar more likely to pay dowry than UP and relate it to social norms. In my understanding, as a reader, both Bihar and UP, two bordering northern states of India have very similar social norms around gender so this argument needs to be validated and cannot be left as an open statement. Does Bihar do worse than UP on indicators of gender equality?

5) I would also suggest estimating separate regressions for rural and urban areas since education effects can vary by region too and it would be good to know how they vary.

Minor comments

6) The authors write – “However, in his study, Anderson refuted the claims of any association between marriage squeeze and dowry payments.” Siwan Anderson is a female economist and therefore the correct pronoun must be used for her.

7) More than legal age is defined by authors as (≤18 years) – seems like a typo?

8) Table 1 – would be good to report the standard errors

6. PLOS authors have the option to publish the peer review history of their article (what does this mean?). If published, this will include your full peer review and any attached files.

Reviewer #1: No

Reviewer #2: No

Reviewer #3: **Yes**

---

## [Author Response · Author response to Decision Letter 0]

8 Jun 2021

Editor’s comments:

 Response: Authors have submitted a file that followed all the required guidelines laid down by jounal.

2. Please amend either the title on the online submission form (via Edit Submission) or the title in the manuscript so that they are identical.

Response: Title has been modified as suggested. 

Reviewer #1: Referee Report on "Banned by the law, practiced by the society: The study of factors associated with dowry payments in Uttar Pradesh and Bihar, India"

Summary & General Evaluation

1. Dowries are wealth transfers from the bride’s family to the groom or groom’s family paid at the time of the wedding. In India, the practice of dowry payment is highly prevalent and typically several times the yearly household income, despite being illegal since 1961. This paper studies the factors associated with a dowry payment using data from the Understanding the Lives of Adolescents and Young Adults (UDAYA) project, consisting of 5206 married adolescent girls from two states in India, Utter Pradesh, and Bihar. Using logistic regression analysis, the authors document associations between the practice of dowry payment and various socio-economic factors such as the age of the girl, education of the girl, household socioeconomic status, spousal education and, spousal age gap, to name a few.

Response: Authors are thankful to the reviewer for reading the manuscript critically and for giving his valuable inputs.

2. Let me start by saying that I find the exercise of establishing correlations between dowry payments and important socio-economic variables worthwhile, and the paper definitely furthers our understanding of the possible determinants of the decision to pay a dowry along with confirming some of the previous findings in the literature. However, in its current version, the paper could be significantly improved in terms analysis and writing. Below I summarize my main comments.

Response: Authors have carried out the revisions as suggested by the reviewer.

Main Comments

3. Data and Regression Analysis. The following are my specific concerns with the data and analysis in the paper (The main text from the article is in blue with quotations): "The sample size for this study was 5,206 adolescent girls who were married at the time of the survey" – The original sample is 10,350 adolescents from Utter Pradesh and 10,350 adolescents from Bihar; from this, you select 5206 adolescents. Please explain the sample selection procedure.

Response: Dear reviewer, we dealt with only married adolescent girls aged 15-19 (N=5206). The remaining sample was for unmarried adolescent boys and girls. We mentioned in the last line of the method section “ The sample size for this study was 5,206 adolescent girls who were married at the time of the survey”

4. "Wealth index was recoded as poorest, poorer, middle, richer, and richest. The survey measured household economic status, using a wealth index composed of household asset data on ownership of selected durable goods, including means of transportation, as well as data on access to a number of amenities. The wealth index was constructed by allocating" - Doing a Principal Component Analysis on the variables and, creating a wealth component would be preferred.

Response: Dear reviewer, we used Principal Component Analysis for creating wealth component. However, it was not mentioned. We have now mentioned that in the respective variable description. 

5. "This is consistent with our study's findings, where the girls from the wealthier family were more likely to pay the dowry at marriage." – I am not sure what variable are you using for natal family wealth?

Response: We used the household wealth index to measure girls’ level of economic status. Wealth index is a proxy indicator to understand the economic status of the household. This is a composite index constructed by using the information such as availability of land, having Radio, TV, Type of house, Source of drinking water, Type of latrine facility etc. 

6. Since your sample by deﬁnition only includes girls aged 15-19. The fraction of girls who are legally going to marry above the legal age of 18 is by construction going to be small. The authors should mention this point in the paper.

Response: Comment incorporated in the description of the respective variable description. 

7. π needs to defined precisely in the specified logistic regression.

Response: comment incorporated. 

8. As a robustness check, the authors could see if these correlations hold true using a linear probability model instead of logistic regression.

Response: Dear reviewer, thank you for such a useful insight. The authors checked for the linear probability model instead of logistic regression; however, as per the literature available and results from our analysis revealed that logistic regression model was best fit model. 

9. Relation to existing works. The authors could have been more thorough in writing this paper, especially in citing relevant literature while explaining the main ﬁndings. There is extensive theoretical literature in economics on the emergence and the existence of dowries. In the introduction, while motivating the presence of dowries in societies like India, the authors cite some of the papers such as Anderson (2003) but miss out on some critical articles like Anderson and Bidner (2015); Botticini and Siow (2003).

Response: the authors are thankful to the reviewer for suggesting some potential studies that the authors missed. Accordingly the introduction section has been revised by including the suggested studies.

10. Similarly, there is a growing empirical literature studying the determinants of dowry payments that have carefully analyzed questions related to this paper’s main ﬁndings. I recommend the authors carefully review the following articles:

Chiplunkar and Weaver (2019) carefully document the transition of dowry payments in India using the 1999 wave of the ARIS-REDS data and test which theories about dowry inﬂation are consistent with the data which are not. I highly recommend the authors to read this paper thoroughly. For example, the authors in the introduction talk about the Sanskritization theory, Chiplunkar and Weaver (2019) show that this theory cannot explain dowry inﬂation. Similarly, they also ﬁnd that the REDS data offers limited support to the marriage squeeze hypothesis.

Edlund (1999): The author also studies the hedonic regressions of dowry on bridal traits. However, she looks at actual magnitude dowry payments, different from this paper that looks at dowry payments on the extensive margin. A couple of sentences comparing results in this paper to yours will be beneﬁcial.

Arunachalam and Logan (2016) is also a related paper.

Response: Authors are very much thankful to the reviewer for critically studying the paper and suggesting relevant literature. Accordingly, we have included all the suggested literatures in the manuscript. 

11. Exposition and takeaways. The discussion and the conclusion section need to significantly re-written for clarity. The authors make a series of claims in the discussion section that require a relevant citation. Similarly, the conclusion section can also be reworded in line with the main contribution of the paper. I list some of the specific instances below (The main text from the article is in blue with quotations):

"Researchers unanimously agreed that the problem lies with gender inequality and female deprivation at every stage" - This sentence needs to be reworded for clarity, and relevant literature that has documented the relationship between dowry payment and gender inequality at different stages of a woman’s life needs to be cited (Alfano, 2017; Bhalotra et al., 2020; Bloch and Rao, 2002; Zhang and Chan, 1999). Further, the claim that dowries are associated with female deprivation at every stage is not supported by the existing literature (Zhang and Chan, 1999).

Response: Authors are thankful to the reviewer for suggesting the previously related study. We have read all the studies coherently and added the related information in the revised manuscript as suggested by the reviewer.

12. "Despite acknowledging the problem of dowry widely, there is a paucity of empirical studies that systematically analyze the correlates of dowry among adolescent girls in recent times." - The authors need to cite relevant papers here.

Response: Authors have cited the relevant literature at the given place in the text. 

13. "However, with the transition of time, the practice of dowry is becoming mandatory, and the demand for a higher dowry becomes a burden to the bride’s family." - There is a shift in dowries from a stridhan to a groom-price (Srinivas, 1984), but what is the evidence that the practice is becoming more mandatory?

Response: The authors want to clarify here, that in the above statement, we mean to say that dowry becomes a common practice, instead of mandatory. We have reframed the sentence

14. "Finally, national representative data will be helpful for further study to understand the scenario of dowry practice in India, as because India is a country with diverse social and cultural practices, dowry will vary with respect to their cultural norms." - Rural Economic and Demographic Survey (REDS) of India is a nationally representative survey that contains dowry information.

Response: I do agree that REDS is a nationally representative data. However, REDS conduct the survey only in Rural India and a sample of only 9500 household will not give good country level representation for a country like India with a large population. 

15. "Instead, a massive social reform and action are urgently required to stop the practice and change their attitude about the system. A community-level approach is necessary to develop their level of understanding and awareness to understand the negative impact of such an evil custom. Simultaneously, it is necessary to restructure the existing dowry prohibition law to make it more effective. There are some unique and exceptional causes regarding dowry that need to be considered during policy development." - This goes well beyond the scope of the paper. The paper is documenting interesting correlations in a unique dataset between dowry payment and socio-economic characteristics. The conclusion should be about these associations.

Response: Thank you for the comment. We now have incorporated the suggestion.

16. Other comments: "However, in his study, Anderson refuted the claims of any association between marriage squeeze and dowry payments [11]." - This sentence has a notable typo; it should be in her study. The paper in its current version has several typos and grammatical errors. The references section needs to be edited, too; please follow one reference style consistently throughout the paper.

Response: Thank you for highlighting the issue of typing error. The same has been corrected. Also, the references have been modified as per given suggestion. 

Reviewer 2:

Summary: This paper looks at the correlates of the prevalence of dowry in UP and Bihar in India. It uses an adolescent survey conducted in these two states. Using a conditional correlation empirical setup, it documents important correlates such as education, wealth, and age at marriage, thereby aiming to contribute to the knowledge of dowry prevalence in India. There are some key areas in which this paper needs significant modifications. Following comments are aimed at highlighting those areas and providing suggestions wherever possible.

Response: Authors are thankful to the reviewer for providing his valuable inputs.

Comments: 

1. While the goal of the paper is clearer, the contribution needs more work. Authors cite literature that has looked at the factors explaining cross-sectional differences in dowry practice and linking it to marriage market conditions and socio-economic factors. The variables authors look at are not too different from these factors and the contribution seems unclear. Authors need to clearly explain how their study is an addition/complementary to the existing work. In addition to that, the paper needs to document what gaps it is attempting to bridge in the literature. While doing that, a clear and coherent discussion of the existing literature is required. It is imperative to document the main findings of existing studies and any unexplored gaps. Finally, how this study aims to fill those gaps and what equips this study to look further than the existing literature. Also, how is the survey different from other larger-scale publicly available survey data such as IHDS and DHS? What is specific about this survey that allows for an exploration which would have not been possible otherwise (if so)?

Response: The manuscript has been revised on the suggested lines. Introduction and discussion section has been improved a lot. 

2. This comment relates to the lack of mechanisms in this paper. The paper relies on existing literature and links their findings to the empirical analysis in this paper. However, this is not adequate in terms of exploring mechanisms. In absence of a non-causal setup, this becomes particularly difficult but no less important. Before linking the correlates to the theoretical work in the field (as cited by the authors), authors should try to explore the mechanisms of their findings using an empirical framework and the data used in this study. 

Response: We have analysed the correlates of dowry in two states of India without much focus on mechanism and empirical framework. However, we have extensively improved upon the introduction section that provides a viable framework for this study.

a) Regional or cluster level heterogeneities in explanatory variables can be utilized to explore the link between the prevalence of dowry and potential mechanisms (which can often be conflicting based on the findings of existing theoretical and empirical literature). These heterogeneities could be based on: 

I. Regional differences in education levels of marriageable girls and boys.

II. Regional difference in age of marriage for both girls and boys.

III. Regional differences in wealth. 

IV. Regional differences in caste mix. Homogeneity in caste mix may have different implications for dowry practice as compared to higher diversity in caste mix working through the channels of societal norms, culture, etc.

Response: A supplementary table is now added with the disaggregated analysis by region. 

b) Dowry prevalence is highly correlated with individual characteristics and regional or group characteristics. It can be thought of as a function of differences in the distribution of individual independent variables and the differences in groups and regions induced by 2 these independent variables. Authors should explore if they can decompose the observed effect into these categories using methods like Oaxaca-Blinder decomposition. This may go a long way in unmasking some important effects and put them in the context of existing literature. 

Response: Thanks for the suggestion. The authors completely agree with this. However, applying Oaxaca-Blinder decomposition is beyond the scope of this paper. The author may explore this in future research on the next round of UDAYA data where a sample of married women would be large enough to do such analysis including intersectionality or combinations of groups, regions and decomposing their effects.

c) After the empirical work exploring the channels that data allows for, existing literature on societal norms, culture, and prevalence of dowry can form a more organic link between the findings and possible explanations of the findings. 

Response: The introduction and discussion section has been updated to incorporated the changes on the suggested lines.

d) Authors can use other publicly available primary data such as DHS and IHDS to explore a much larger set of regional heterogeneities in relevant explanatory factors. This however is condition on being able to link the regions (district etc.) in the survey to these other datasets.

Response: Authors are aware of those data source, however, the aim was to cetagorize the risk factor of dowry in two of the most backward states in India utilizing the latest data source. 

3. Empirical work in this study needs improvement on multiple grounds. Some very important areas are below: 

a) The paper currently lacks the features of a representative conditional correlational analysis in the field. In the presence of a potential omitted variable bias and no direct measure in form of a causal framework to counter that, I would suggest the following:

 I. Testing the sensitivity of explanatory variables to addition and removal of other linked controls. All explanatory factors should be added sequentially and their sensitivity to the addition of further variables/controls should be documented before the direction and magnitude of their correlation to dowry prevalence can be discussed in results. For a few key variables such as household wealth, education, etc – their coefficient stability can be subjected to randomly dropping other regional and demographic controls.

Response: Authors have tried the analysis as suggested by the reviewer. Accordingly, three supplementary tables have been added during the revision. Certain analysis were not possible as they were not related to study objectives, or were not significant or authors were not aware of the techniques.

 II. Explore the coefficient stability more formally using statistical exercises that provide bounds for the treatment effects, such as Oster (2016).

Response: Dear reviewer, this is for the linear model, authors are not sure in the case of logistic if it can be applied

b) Multiple correlates can be thought of as multiple hypotheses being tested. The coefficients should be subjected to tests for multiple hypotheses such as BenjaminiHochberg correction. 

Response: Dear reviewer the authors are not aware of this technique. As per authors knowledge, cross tabulations between multiple variables (with categorical nature) and using chi square test is sufficient to establish the association. 

c) Discuss why a logistic distribution is more appropriate for this setup as compared to something like a linear probability model. 

Response: Dear reviewer, thank you for such a useful insight. The authors checked for the linear probability model instead of logistic regression; however, as per the literature available and results from our analysis revealed that logistic regression model was best fit model. 

d) Formally describe the model along with the variable construction. Are there any regional fixed effects? What is the level of clustering of standard errors?

Response: We have analysed the data and results are presented in supplementary tables. We have made analysis urban-rural wise to depict regional variations.

 e) Results should be reported with the difference in coefficients and standard errors.

Response: Dear reviewer, in table-2 the authors have added standard errors. 

 f) Are there any sampling weights? They are more important in conditional correlations.

Response: Dear reviewer, the authors used individual weights to provide representative estimates. Now added in the statistical analysis section. 

Reviewer 3:

Reviewer #3: The paper uses data on 5206 married adolescent girls from the Understanding the lives of adolescents and young adults (UDAYA) conducted in Uttar Pradesh and Bihar for studying correlates of dowry payment in India. Main findings are - dowry likelihood lower if husband is known to the female, if the adolescent’s mother has more than 10 years of education; dowry likelihood higher if the couple is more educated, girl above legal age, husband was older, wealthier families, and in rural areas.

Major comments

1) Definition of dowry and who answered the question matters: While the authors mention that their main variable of interest comes from what a household’s response is to the question – “whether dowry paid at the time of marriage or later?” – what is not clear is the inclusions in the term “dowry”. A clarity on this would be valuable for the readers. There are two potential measurement issues with this variable

Response: A more refined discussion on dowry has been included to avoid any confusion.

a) One, if perception of dowry (inclusions) and hence reporting varies by education or other economic correlates, then this can potentially contaminate the findings of the paper.

Response: Authors have revised the paper on the suggested lines.

b) Second, dowry is a sensitive issue and reporting might vary – although whether it varies along the dimensions that the authors study would need to be argues. If it is underreported by the same fraction by all groups then it does not matter. However, the rural-urban differential can vary because of reporting sensitivity too where urban households maybe aware of dowry prohibition act.

Response: The reporting of dowry could be underreported and authors agreed with this point. Furthermore, we have provide supplementary table to look into the urban-rural differential as suggested by the reviewer.

c) Importantly, who answers the question is also important and implications of these both should be adequately discussed, even though I suppose addressing these issues is not feasible.

Response: A more refined discussion has been included now.

2) The authors mention – “Higher education is often found to be associated with higher dowry; this is because due to the competition among the brides for a particular groom leads to offers of higher and higher dowries” and cite a paper by Munshi (2017). But I am not certain if this is the only theoretical channel that should drive the effect of education on dowries. It is possible that more education leads to more awareness about evils of dowry and can potentially also lower the dowry payment? In general, it would be good to present a theoretical framework of why each result that the authors obtain can be justified theoretically (not a theoretical model, but channels are enough). At least, what are the hypothesis of the authors (based on different channels) should be mentioned before they go on to the empirical strategy and discuss the covariates they include. The current discussion in the results seems a bit superfluous and lacks conceptual clarity, hence the results do not seem to come together coherently.

Response: Thank you for the comment. We now have incorporated the comment. 

3) While discussing the results the authors discuss results from previous studies which examine both probability of dowry payment and amount of dowry paid – it would be good for a reader to clearly differentiate between these two types of studies.

Response: Thank you for the comment. We have incorporated the comment and cited the source of the study. 

4) The authors also make an argument on page 13 about girls from Bihar more likely to pay dowry than UP and relate it to social norms. In my understanding, as a reader, both Bihar and UP, two bordering northern states of India have very similar social norms around gender so this argument needs to be validated and cannot be left as an open statement. Does Bihar do worse than UP on indicators of gender equality?

Response: Thank you for the comment. We now have reframed the discussion for this finding.

5) I would also suggest estimating separate regressions for rural and urban areas since education effects can vary by region too and it would be good to know how they vary.

Response: Comment incorporated and results are depicted through supplementary tables.

Minor comments

6) The authors write – “However, in his study, Anderson refuted the claims of any association between marriage squeeze and dowry payments.” Siwan Anderson is a female economist and therefore the correct pronoun must be used for her.

Response: The authors are really sorry for typing error. The same comment has also been raised by another reviewer. Accordingly, the error has been corrected.

7) More than legal age is defined by authors as (≤18 years) – seems like a typo?

Response: Comment incorporated. 

8) Table 1 – would be good to report the standard errors

Response: Dear sir, in table-1 the authors estimated descriptive statistics using percentage and sample. So, it was not possible to add standard errors in that table. However, in table-2 we have provided standard errors for the estimates. 

---

## [Editor Report · Decision Letter 1]

14 Jun 2021

PONE-D-21-04546R1

Banned by the law, practiced by the society: The study of factors associated with dowry payments among adolescent girls in Uttar Pradesh and Bihar, India

PLOS ONE

Dear Dr. Kumar,

Thank you for submitting your manuscript to PLOS ONE. I enjoyed reading the revised draft. The paper has clearly improved from the last version, but I see that you have not carried out many revisions that I was hoping. I will like you to address the following:

1. Elaborate on the sample selection procedure – what steps were followed.

2. Robustness on LPM must be added as a table.

3. Please use Oster (2016) when you do the LPM (linear probability model).

4. I will like to see a table comparing the basic descriptive statistics with IHDS or REDS (I will prefer REDS over IHDS).

After you resubmit the paper I will handle the decision myself. To make this easier, please send me a letter telling me which changes you made.

Thank you.

Nishith

We look forward to receiving your revised manuscript.

Kind regards,

Nishith Prakash, Ph.D.

Academic Editor

PLOS ONE
---

## [Author Response · Author response to Decision Letter 1]

18 Jun 2021

1. Elaborate on the sample selection procedure – what steps were followed.

Response: Comment incorporated (Figure-1).

2. Robustness on LPM must be added as a table.

Response: Comment incorporated. 

3. Please use Oster (2016) when you do the LPM (linear probability model).

Response: Comment incorporated.

4. I will like to see a table comparing the basic descriptive statistics with IHDS or REDS (I will prefer REDS over IHDS).

Response: The authors cannot use IHDS or REDS data for the following issues.

(a): IHDS data gives estimates at national level and therefore, state-level analysis cannot be performed. It is not a good idea to perform state-level analysis as the data is not state-representative. In this study, we have used UDAYA data specifically designed for Uttar Pradesh and Bihar. Conceptually, it is not feasible to compare the two datasets.

(b): The UDAYA data we used is from 2016. However, the latest REDS data is from 2006. Therefore, the descriptive analysis may not be that useful. So, authors feel that it won’t be a good idea to generate a table (using IHDS/REDS data) for bivariate association.

---

## [Editor Report · Decision Letter 2]

28 Jun 2021

PONE-D-21-04546R2

Banned by the law, practiced by the society: The study of factors associated with dowry payments among adolescent girls in Uttar Pradesh and Bihar, India

PLOS ONE

Dear Dr. Kumar

There are few things I really want you to address:

1. I cannot find the Oster (2017) table.

2. Please have detailed notes in all tables -- some have no notes.

3. Have proper labels for all variables used in the tables.

4. IHDS (2005 and 2011) can be used for state level analysis as its representative (so is REDS). Although its an older data, it important to show the table.

I cannot proceed unless you address these 4 points. Also, I found the tables quite sloppy.

Best,

Nishith

We look forward to receiving your revised manuscript.

Kind regards,

Nishith Prakash, Ph.D.

Academic Editor

PLOS ONE
---

## [Author Response · Author response to Decision Letter 2]

20 Jul 2021

Editor’s comments:

1. I cannot find the Oster (2017) table.

Response: Dear Editor, We are really sorry as we are unable to understand the Oster (2017) table. We tried our best but could not prepare the required table. We also took help from some of the individuals/researchers, however, none of them could come out to our rescue as everyone failed to understand the concept. We are really sorry on our part as we were unable to understand the relevant methodology and therefore, we are not including Oster (2017) table. If the table is absolute necessity, we would seek the editor’s guidance in this regard.

2. Please have detailed notes in all tables -- some have no notes.

Response: All the tables have been updated as suggested.

3. Have proper labels for all variables used in the tables.

Response: Labels have been formatted as per the suggestion.

4. IHDS (2005 and 2011) can be used for state level analysis as its representative (so is REDS). Although its an older data, it important to show the table.

Response: Authors have used IHDS data to show the relevant information. Please find the table attached to this response. 

Table : Socio-economic characteristics of married girls 15-19, 2004-05 and 2011-12

Background characteristics IHDS 2004-05 IHDS II 2011-12

 Sample Percentage Sample Percentage 

Gift paid at the time of marriage 

No 0 0 N/A N/A

Yes 347 100 N/A N/A

Age at marriage (in years) 

Less than 15 124 35.7 70 30.0

15 and more 223 64.3 163 70.0

Literate 

No 196 57.0 79 33.9

Yes 148 43.0 154 66.1

Working status 

No 256 73.78 184 79.0

Yes 91 26.22 49 21.0

Caste 

SC/ST 95 27.4 67 28.8

Non-SC/ST 252 72.6 166 71.2

Religion 

Hindu 293 84.4 184 79.0

Non-Hindu 54 15.6 49 21.0

Wealth index 

Poorest 97 28.0 47 20.2

Poorer 61 17.6 47 20.2

Middle 77 22.2 46 19.7

Richer 59 17.0 47 20.2

Richest 53 15.3 46 19.7

Residence 

Urban 66 19.0 37 15.9

Rural 281 81.0 196 84.1

State 

Uttar Pradesh 245 70.6 155 66.5

Bihar 102 29.4 78 33.5

Total 347 100 233 100

N/A: Data Not available; 

Dowry is assessed from the question "Generally in your community for a family like yours, what are the kind of things that are given as gifts at the time of the daughter's marriage?" Gold, Silver, Land, Car, Scooter/motorcycle, TV, Fridge, Furniture, Pressure Cooker, Utensils, Mixer grinder, Bedding/mattress, Watch, Bicycle, Sewing machine, Livestock, Tractor and Cash.

* Please note that the dowry prevalence is 100 percent for the year 2004-05. The question related to dowry asked in IHDS was different than what was asked in UDAYA data. 

Furthermore, authors could not utilize REDS data as the relevant data were not provided to the authors. We requested to the concerned authority for data access, however, we were told that they only grant access in certain conditions that are needed to be fulfilled by the institution. They only grant access to the institution and from institution only we can borrow data after receiving proper IRB approval which was not feasible in this case.

This is what they said:

To obtain cross-walk to village identifiers and/or previous survey rounds or any of the 2006 data, the investigator must obtain IRB approval from their respective university or institution. The IRB-approved proposal should indicate the following:

a) that statistical identification of households or villages is possible with the secure data

b) that linked data and cross-walks will be kept in a secure environment

c) names of all investigators and research assistants with access to the linked data

d) that linked data will only be used on projects with IRB approval 

e) that the linked data or the cross-walk cannot be provided to investigators or researchers 

not specifically identified in (c) 

f) that publications and presentations must not reveal village or individual level identifiers. 

I cannot proceed unless you address these 4 points. Also, I found the tables quite sloppy.

Response: We have edited the tables as per given suggestions. Furthermore, we tried to incorporate all the comments suggested by the editor except that Oster (2017) related comment. We seek advice from editor on this issue.

---

## [Editor Report · Decision Letter 3]

17 Aug 2021

PONE-D-21-04546R3

Banned by the law, practiced by the society: The study of factors associated with dowry payments among adolescent girls in Uttar Pradesh and Bihar, India

PLOS ONE

Dear Dr. Kumar,

Thank you for submitting your manuscript to PLOS ONE. After careful consideration, we feel that it has merit but does not fully meet PLOS ONE’s publication criteria as it currently stands. Therefore, we invite you to submit a revised version of the manuscript that addresses the points raised during the review process.

We look forward to receiving your revised manuscript.

Kind regards,

Nishith Prakash, Ph.D.

Academic Editor

PLOS ONE

Journal Requirements:

Additional Editor Comments (if provided):

Please see: https://emilyoster.net/research/. On this page look for JOURNAL OF BUSINESS ECONOMICS AND STATISTICS, JUNE, 2019, Unobservable Selection and Coefficient Stability: Theory and Validation. They provide the do file to undertake the exercise. W/o this paper I cannot proceed.

Thank you.

Nishith
---

## [Author Response · Author response to Decision Letter 3]

30 Sep 2021

MANUSCRIPT TITLE: Banned by the law, practiced by the society: The study of factors associated with dowry payments among adolescent girls in Uttar Pradesh and Bihar, India

MANUSCRIPT ID: PONE-D-21-04546R3

Respected editor, 

Thank you for giving us the opportunity of submitting an improved version of our manuscript for publication in the PLOS One. We are highly grateful to receive your insightful comments and suggestions. We appreciate the time and effort that you have put forward to provide valuable feedback that has significantly improved our paper. Kindly note that we have incorporated the changes that were suggested. The modifications have been shown using track changes within the revised manuscript. Please see below for a point-by-point response to each of the comments and suggestions.

Hoping that you and your family members are safe and sound in these challenging times,

Yours Sincerely,

Authors

 

EDITOR COMMENTS:

1. Journal Requirements: Please review your reference list to ensure that it is complete and correct. If you have cited papers that have been retracted, please include the rationale for doing so in the manuscript text, or remove these references and replace them with relevant current references. Any changes to the reference list should be mentioned in the rebuttal letter that accompanies your revised manuscript. If you need to cite a retracted article, indicate the article’s retracted status in the References list and also include a citation and full reference for the retraction notice.

Response: Thank you for pointing this out. Comment has been incorporated.

2. Additional Editor Comments (if provided): Please see: https://emilyoster.net/research/. On this page look for JOURNAL OF BUSINESS ECONOMICS AND STATISTICS, JUNE, 2019, Unobservable Selection and Coefficient Stability: Theory and Validation. They provide the do file to undertake the exercise. W/o this paper I cannot proceed.

Response: Dear Editor thank you very much for the suggestion. Based on your comment, we proceeded for stability check of the study coefficients after reading Emily Oster’s research article. The results for the same has been shown in Table 4 of the manuscript. The methodology has been discussed in the “statistical methods” section as shown below:

Next, we check the stability of the regression coefficients and their sensitivity to selection bias using standard methods. We obtain bias-adjusted coefficients and calculate the absolute deviation from the non-bias-adjusted regression estimates to understand the extent of bias. Further, we calculate Oster's δ, whose value higher than one would indicate that the regression coefficients are insensitive to omitted variable bias and variable selection bias. All estimates were obtained with the assumption that the bias-adjusted model would explain 1.3 times variation in dowry payment status compared to the non-bias-adjusted model. The statistical analyses for coefficient stability check were performed using the psacalc command by estimating linear probability models in STATA.

The results for the stability check are discussed in the “Results” section (Page 13):

Table 4 gives the results of the coefficient stability check of the explanatory variables of dowry payment among female adolescents. From the bias-adjusted estimates (see column 8), we observe that the multivariable association between husband familiar before marriage, age at marriage, spousal education, wealth index, residence and state with dowry payment is statically significant (at 5% level) and lies in the same direction as the uncontrolled estimates. Moreover, from the difference shown in column 10, we can say that the bias-adjusted and non-bias-adjusted regression coefficients are similar. However, Oster's delta revealed that the statistically significant multivariable association of age at marriage and state with dowry payment suffers from omitted-variable and selection bias.

Further, we have added the following line as study limitation in the discussion section (Page 18):

Although the coefficient stability check revealed that the majority of the explanatory characteristics are insensitive to omitted-variable and selection bias, the results for age at marriage and state need to be interpreted with caution.

---

## [Editor Report · Decision Letter 4]

4 Oct 2021

Banned by the law, practiced by the society: The study of factors associated with dowry payments among adolescent girls in Uttar Pradesh and Bihar, India

PONE-D-21-04546R4

Dear Dr. Kumar,

We’re pleased to inform you that your manuscript has been judged scientifically suitable for publication and will be formally accepted for publication once it meets all outstanding technical requirements.

Kind regards,

Nishith Prakash, Ph.D.

Academic Editor

PLOS ONE

Additional Editor Comments (optional):

Dear Dr. Kumar

Very happy to accept this paper. Congratulations!

Best,

Nishith
---

## [Editor Report · Acceptance letter]

8 Oct 2021

PONE-D-21-04546R4 

Banned by the law, practiced by the society: The study of factors associated with dowry payments among adolescent girls in Uttar Pradesh and Bihar, India 

Dear Dr. Kumar:

I'm pleased to inform you that your manuscript has been deemed suitable for publication in PLOS ONE. Congratulations! Your manuscript is now with our production department. 

Kind regards, 

on behalf of

Dr. Nishith Prakash 

Academic Editor

PLOS ONE